# Safety of Post-Transplant Cyclophosphamide-Based Prophylaxis in AML Patients with Pre-Existing Cardiac Morbidity Undergoing Allogeneic Hematopoietic Cell Transplantation

**DOI:** 10.3390/cancers17193128

**Published:** 2025-09-26

**Authors:** Arnau Torrent-Rodríguez, Enric Cascos, Víctor Navarro Garcés, Estefanía Pérez-López, Mónica Baile-González, Carlos Martín Rodríguez, María Jesús Pascual Cascón, Marta Luque, Albert Esquirol, Carmen Martín Calvo, Felipe Peña-Muñoz, Inmaculada Heras Fernando, Itziar Oiartzabal Ormtegi, Adolfo Jesús Sáez Marín, Sara Fernández-Luis, Juan José Domínguez-García, Sara Villar Fernández, José Luis López Lorenzo, Miguel Fernández de Sanmamed Girón, Leslie González Pinedo, Lucía García-Maño, Ana Pilar González-Rodriguez, Tamara Torrado, Silvia Filaferro, Pascual Basalobre, Guillermo Ortí, Montserrat Rovira, Manuel Jurado Chacón, María Queralt Salas

**Affiliations:** 1Hospital Clínic de Barcelona, 08036 Barcelona, Spain; atorrentr@clinic.cat (A.T.-R.); cascos@clinic.cat (E.C.); mrovira@clinic.cat (M.R.); 2Faculty of Medicine and Health Sciences, Universitat de Barcelona, 08907 Barcelona, Spain; 3Statistics Unit, Vall d’Hebron Institute of Oncology (VHIO), 08035 Barcelona, Spain; victornavarro@vhio.net; 4Hematology Department, Complejo Asistencial Universitario de Salamanca/IBSAL, 37007 Salamanca, Spain; eperezl@saludcastillayleon.es (E.P.-L.); mbaile@saludcastillayleon.es (M.B.-G.); cmartinrodr.ibsal@saludcastillayleon.es (C.M.R.); 5Hematology Department, Hospital Regional Universitario de Málaga, 29010 Málaga, Spain; mjcascon@gmail.com (M.J.P.C.); martaluque2008@gmail.com (M.L.); 6Hematology Department, Hospital de la Santa Creu i Sant Pau, 08025 Barcelona, Spain; aesquirol@santpau.cat; 7Hematology Department, Hospital Reina Sofía de Córdoba, 14004 Córdoba, Spain; carmen.martin.calvo.sspa@juntadeandalucia.es; 8Institut Català d’Oncologia, Hospital Duran i Reynals, 08907 Barcelona, Spain; apena@iconcologia.net; 9Hematology Division, Hospital Morales Meseguer, 30008 Murcia, Spain; inmheras@um.es; 10Hematology Department, Hospital Universitario Donostia, 20080 Donostia, Spain; itziar.oiartzabalormategi@osakidetza.eus; 11Hematology Department, Hospital Universitario 12 de Octubre, 28041 Madrid, Spain; adolfojesus.saez@salud.madrid.org; 12Hematology Department, Hospital Universitario Marqués de Valdecilla, 39008 Santander, Spain; sara.fernandezl@scsalud.es (S.F.-L.); juanjose.dominguez@scsalud.es (J.J.D.-G.); 13Hematology Department, Clínica Universidad de Navarra, 31008 Pamplona, Spain; svillar@unav.es; 14Hematology Department, Fundación Jiménez Díaz, 28040 Madrid, Spain; jllopez@quironsalud.es; 15Hematology Department, Hospital Universitario de Gran Canaria Doctor Negrín, 35010 Gran Canaria, Spain; mfergirw@gobiernodecanarias.org (M.F.d.S.G.); lgonpin@gobiernodecanarias.org (L.G.P.); 16Hematology Department, Hospital Universitatio Son Espases, 07120 Palma de Mallorca, Spain; lucia.garcia@ssib.es; 17Hematology Department, Hospital Universitario Central de Asturias, 33011 Oviedo, Spain; anapilar.gonzalez@sespa.es; 18Hematology Department, Hospital Universitario de A Coruña, 15006 A Coruña, Spain; tamara.torrado.chedas@sergas.es; 19Grupo Español de Trasplante de Progenitores Hematopoyéticos y Terapia Celular, 28040 Madrid, Spain; sfilaferro@geth.es (S.F.); gorti@vhio.net (G.O.); manuel.jurado.sspa@juntadeandalucia.es (M.J.C.); 20Hematology Department, Hospital Universitario Valle de Hebrón, 08035 Barcelona, Spain; pbalsalobre@geth.es; 21Hematology Department, Hospital Universitario Virgen de las Nieves de Granada, 18014 Granada, Spain; 22Hematopoietic Cell Transplantation Unit, Hematology Department, Institute of Cancer and Blood Diseases (ICAMS), Hospital Clínic de Barcelona, 08036 Barcelona, Spain

**Keywords:** acute mieloid leukemia, cardio-oncology, cardiotoxicity, post-transplant cyclophosphamide

## Abstract

This retrospective study of 461 acute myeloid leukemia patients, including 62 with pre-existing cardiac morbidity, evaluated the safety of post-transplant cyclophosphamide (PTCy). Cardiac events occurred in 13.2%, with no significant differences in incidence, survival, or mortality, supporting PTCy as a safe and feasible option in eligible allo-HCT candidates with appropriate cardio-oncology follow-up.

## 1. Introduction

Post-transplant cyclophosphamide (PTCY) has emerged as a cornerstone strategy in graft-versus-host disease (GVHD) prophylaxis following allogeneic hematopoietic cell transplantation (allo-HCT), due to its effectiveness in reducing GVHD incidence across donor types [1,2,3,4]. Initially developed for use in haploidentical transplantation (haplo-HCT) [1], the PTCY-based approaches have progressively been adopted by transplant centers worldwide, extending its application to matched sibling and unrelated donor settings with considerable success [2,3,4]. Although its immunomodulatory benefits are well documented, the expanding use of PTCY has also raised important questions regarding its safety profile, particularly concerning potential cardiotoxic effects in high-risk patient populations [5,6,7].

Cyclophosphamide (Cy)-associated cardiotoxicity, especially when administered at high doses, is a well-documented clinical toxicity induced by this alkylating agent [8,9]. This risk is further amplified in allo-HCT recipients, many of whom have prior exposure to cardiotoxic agents such as anthracyclines and face important immunomodulatory, inflammatory and endothelial dysregulation changes during the transplant process evolution [5,6,7,10]. In light of these concerns, and given the widespread use of PTCy-based prophylaxis, the Grupo Español de Trasplante Hematopoyético y Terapia Celular (GETH-TC) initiated retrospective multicenter study to investigate the potential association between PTCy administration and the development of cardiac events (CE), in patients with acute myeloid leukemia (AML) previous treated with anthracycline induction treatments undergoing allo-HCT with PTCy [11,12].

Notably, despite the potential cardiotoxic-related risk of Cy, allo-HCT with PTCy is performed in patients with pre-existing cardiac morbidity. In routine clinical practice, these patients are typically evaluated pre-transplant by specialized cardiology teams to assess and mitigate potential risks. Nevertheless, this subgroup represents a clinically relevant population often excluded or underrepresented from prospective trials due to perceived higher vulnerability. Based on this context, this study specifically explores the safety and feasibility of PTCy in patients with pre-transplant cardiac morbidity, and compares the incidence of CE and outcomes with those of patients undergoing allo-HCT without baseline cardiac disease.

## 2. Methods

### 2.1. Study Design and Patient Selection

This is a retrospective and multicenter study conducted by the GETH-TC, a non-profit scientific society representing all hospital units performing HCT in Spain and Portugal. A total of 17 institutions contributed to the project. Patient selection included adults with AML treated with at least one cycle of anthracyclines-based induction therapies undergoing their first peripheral blood allo-HCT with standard doses of PTCY-based prophylaxis from 2012 and 2022. All patients included signed informed consent for retrospective clinical data collection before allo-HCT. The information needed for the conduction of the present study was collected in June 2024. The study was approved by the Ethics Committee of the Hospital Clínic de Barcelona and the GETH-TC, and conducted following the standards set forth by the Declaration of Helsinki.

### 2.2. Prior Cardiac Morbidity, Pre-Transplant Cardiac Evaluations and Main Definitions

Pre-existing cardiac comorbidities (PCC) was defined as any documented cardiac condition diagnosed prior to allo-HCT, based on patients’ clinical charts. As part of the pre-transplant evaluation, all centers systematically performed transthoracic echocardiography (ECHO) and electrocardiograms (ECG). Patients without a prior cardiac history but with an LVEF below 45% were also classified as having pre-existing cardiac morbidity. In general, individuals with known cardiac disease underwent specialized cardiology evaluation before transplantation in all participating centers. Since pre-transplant cardiac biomarkers were not homogeneously recorded, that information was not included in the study. Given the small number of patients with each specific cardiac condition, subgroup analyses were not feasible, as separating them into independent categories would result in very limited statistical power. For this reason, all patients fulfilling these criteria were analyzed collectively as PCC.

A cardiac event (CE) was defined following the methodology defined in previous publications, and as any new occurrence of atrial or ventricular arrhythmias, heart failure (including a ≥10% reduction in LVEF with a final value < 53%), myocardial infarction or ischemia, or pericardial complications diagnosed after the stem cell infusion [5,6,7]. These events were also defined and recorded according to patient medical histories documented in clinical charts. De novo hypertension (HTN) diagnosed after allo-HCT was not considered a CE. CE were further classified as early (ECE), occurring within 100 days post-transplant, or late (LCE), occurring thereafter [13]. All cardiac complications were categorized in accordance with standard clinical practice guidelines and the 2022 European Society of Cardiology (ESC) cardio-oncology guidelines [14]. The severity of cardiac events was graded using the National Cancer Institute Common Terminology Criteria for Adverse Events (CTCAE), version 5.0.

### 2.3. Allo-HCT Information and Main Definitions

Induction therapies, criteria for allo-HCT eligibility, donor selection, and conditioning regimens were applied in line with standard institutional protocols. The intensity of conditioning was typically adapted according to patients’ chronological age and existing comorbidities. All recipients were administered PTCy at 50 mg/kg/day on days +3 and +4. Peripheral blood stem cells (PBSCs) were used as the graft source in all cases. Grading of acute and chronic GVHD (aGVHD and cGVHD) followed established criteria [15,16,17]. Neutrophil engraftment was defined as achieving an absolute neutrophil count exceeding 0.5 × 10^9^/L for at least three consecutive days, while platelet engraftment was considered as maintaining platelet counts above 20 × 10^9^/L for seven days without the need for transfusions. The definitions of complete remission (CR) and relapse were determined by the treating physicians.

### 2.4. Endpoint Definition

The primary endpoint was the occurrence of a CE, after transplantation. Secondary endpoints were non-relapse mortality (NRM), defined as death without prior disease relapse since transplantation; relapse-free survival (RFS), defined as the time from transplantation to relapse or death from any cause; overall survival (OS), defined as the time from transplantation to death from any cause; LCE, defined as a CE occurring more than transplantation days after 100; and ECE, defined as a CE occurring within the first 100 days after transplantation.

### 2.5. Statistical Analysis

Baseline characteristics of the PCC and non-PCC cohorts were summarized descriptively. Categorical variables were reported as frequencies and percentages, and continuous variables as medians with interquartile ranges (IQRs). Baseline group comparisons were performed using Pearson’s χ^2^ test for categorical variables and the Wilcoxon rank-sum test for continuous variables.

Overall survival and RFS were estimated using Kaplan–Meier curves and compared with the log-rank test. Hazard ratios (HRs) were obtained from Cox proportional hazards models.

For outcomes affected by competing risks—NRM, CE, ECE, and LCE—cumulative incidence functions (CIFs) were used. A competing risk was defined as an event that prevents the occurrence of the primary event of interest (e.g., in CE/ECE/LCE analyses, death or relapse without a cardiac event; in NRM analyses, relapse). In addition to the main competing risk models, we performed cause-specific hazard models distinguishing deaths directly attributable to cardiac events from deaths due to other causes (Appendix A). Comparisons between groups were performed using Fine–Gray subdistribution hazard models, which directly estimate covariate effects on the CIF while accounting for competing events.

To address baseline imbalances, two propensity score–based methods were applied: (1) 4:1 propensity score matching (PSM) and (2) inverse probability weighting (IPW) using the average treatment effect among the treated (ATT) approach. Both approaches adjusted for key prognostic covariates (age, HTN, conditioning intensity, disease status, donor type, HCT, and dislipidemia), achieving standardized mean differences (SMDs) <0.1 across all variables, indicating adequate balance (Appendix A).

For IPW analyses of competing risk endpoints, direct Fine–Gray modeling was not applicable. Instead, confounder-adjusted CIF curves were estimated using inverse probability weighting, following the methodology described by Denz et al. [18]. Group differences were then assessed by comparing CIF ratios at pre-specified time points (24 and 36 months for all endpoints except ECE, for which 1 and 3 months were used).

Missing data in the covariates used for the IPW model were minimal (<5% for all variables, see Appendix A). Missingness was assumed to be completely at random. We applied multivariate imputation by chained equations (MICE), a flexible approach that handles both continuous and categorical variables. Estimates obtained from imputed datasets were consistent with complete case analyses. All analyses were conducted in R version 4.2.2 [19].

## 3. Results

### 3.1. Patients Characteristics and Allo-HCT Information

A total of 461 AML patients were included in the study. As reported in Table 1, the median age was 55 years (IQR 43–63), and 54.9% were male. PCC were present in 62 (13.4%) patients, with valvopathy (33.8%), arrhythmias (27.4%), heart failure (19.3%), ischemic heart disease (6.2%), and pericardial disease (6.2%) being the most prevalent ones.

As shown in Table 1, baseline characteristics of patients according to PCC were compared, showing that PCC patients were significantly older (median 60 vs. 54 years, *p* = 0.016) and more frequently had HTN (31% vs. 19%, *p* = 0.05). In contrast, the proportion of males and females was similar between groups (*p* = 0.12), and other cardiovascular risk factors such as dyslipidemia (15% vs. 13%, *p* = 0.9), diabetes mellitus (8% vs. 9%, *p* = 0.9), and obesity (8% vs. 7%, *p* > 0.9) were also comparable between groups.

Overall, 249 (54%) patients underwent MAC allo-HCT and 30 (6.5%) received grafts from MMUD and 268 (58%) from haploidentical donors. Transplant practices and donor selection were similar between groups except for the proportion of patients receiving MAC regimens, which were used more frequently in non-PCC (40% vs. 56%, *p* = 0.029) (Table 1).

Baseline characteristics before and after adjustment using IPW and PSM are shown in Table 2.

### 3.2. Cumulative Incidence of Early and Late Cardiac Events

Overall, 61 (13.2%) patients had cardiac complications after allo-HCT. Among these, 35 (57.4%) events were classified as ECE, while 26 (42.6%) were identified as LCE. Interestingly, although a higher proportion of CE was observed in patients with PCC, the difference was not statistically significant when compared to patients without (3-year cumulative incidence of CE: 21% vs. 12%, respectively; *p* = 0.09).

As shown in Figure 1 and Table 3, among patients with PCC, ECE occurred in 7 (11.3%) out of 62 patients, with a median onset at 7 days (IQR 3–29) after allo-HCT, and a day +100 cumulative incidence of 11% (95% CI: 4.9–21%). Among these patients, the most common pre-transplant morbidities were arrhythmia (42.8%) and heart failure (28.5%). Of the 7 ECE cases in this group, 4 (57.1%) were newly diagnosed cardiac complications, while 3 (42.9%) were considered worsening of known pre-existing conditions, as the event matched the patient’s prior cardiac diagnosis. In contrast, among the 399 patients non-PCC, ECE occurred in 28 (7.0%) patients, with a median onset at 9 days (range: 4–49 days) (*p* = 0.933), and a 100-day cumulative incidence of 7.0% (95% CI: 4.8–9.9%).

The types of ECEs were similar between both groups (*p* = 0.528). According to Table 3, arrhythmias and heart failure were the most prevalent ECE. Heart failure occurred in 42.9% of patients with PCC and 21.4% of those without. Arrhythmias were reported in 42.9% and 28.6% of patients, respectively. Pericardial complications were observed in 14.3% of patients with PCC and 35.7% of those without, and ischemic events were only reported in patients non-PCC (10.7%). According to CTCAE severity, grade 3–4 ECE occurred in 2 (28.6%) patients with PCC and in 10 (35.7%) patients without (*p* = 0.059). Moreover, 2 (28.6%) patients with PCC and 2 (7.1%) patients without died directly due to ECE (grade 5) (*p* = 0.59).

LCE are also reported in Table 3. Among patients with PCC, LCE occurred in 5 (8%) out of 62 patients, with a median onset at 469 days (IQR 264–574) after allo-HCT, and a 2-year cumulative incidence of 12% (95% CI: 4.4–25%). The most common pre-existing comorbidities among those who developed LCE were valvular heart disease (60.0%) and arrhythmia (20.0%). Interestingly, all LCEs in this group represented newly diagnosed events. In patients non-PCC, LCE occurred in 21 (5.3%) out of 399 patients, with a median onset at 426 days (IQR 157–708) ant (*p* = 0.855), and a 2-year cumulative incidence of 2.3% (95% CI: 1.4–5.3%).

The distribution of LCE types did not differ between groups (*p* = 0.111). As shown in Table 3, heart failure was the most frequent complication, observed in 60.0% patients with PCC vs. 33.3% patients without. Arrhythmias were reported in 20.0% and 9.5% of patients, while pericardial complications occurred in 20.0% and 9.5% of patients. Grade 3–4 LCE occurred in 0 patients with PCC and in 6 (28.6%) patients without. In addition, 2 (9.5%) patients died due to the LCE, and both of them were classified into the non-PCC group.

Complementary analyses with PSM and IPW confirmed these findings. After adjustment, no significant differences were observed in the incidence of ECE, LCE, overall cardiovascular events, NRM, or OS between patients with and without pre-transplant cardiac morbidity (Figure 1).

### 3.3. Efficacy Outcomes

Despite the difference on pre-transplant comorbidity burden between groups, post-transplant outcomes were globally comparable in between. As reported in Table 1, 2-year OS was 69% (95% CI: 64–73%) in patients with PCC vs. 70% (95% CI: 60–83%) in those without (*p* = 0.50), while NRM was 18% (95% CI: 9.5–29%) vs. 17% (95% CI: 13–21%), respectively (*p* = 0.20).

Interestingly, among patients who developed ECE, those with PCC had worse outcomes, with a 2-year OS rate of 28.6% (95% CI: 4.1–61.2%) compared to 59.9% (95% CI: 39.3–75.4%) in those without (*p* = 0.064). In addition, NRM rates were also different between groups, with a 2-year NRM of 57.1% (95% CI: 12.1–86.2%) vs. 32.1% (95% CI: 15.8–49.8)*,* respectively (*p* = 0.255).

Consistently, adjusted analyses with PSM and IPW did not reveal significant differences in OS or NRM between groups (Figure 2).

Among patients who experienced an ECE, 5 of 7 in the PCC group (71%) and 2 of 28 in the non-PCC group (7%) died directly due to the cardiac event. Although numbers were small and the difference did not reach statistical significance, this suggests a higher vulnerability among patients with pre-existing cardiac morbidity. To further address this, we performed cause-specific hazard models distinguishing CE-related deaths from deaths due to other causes (Appendix A), no significant differences were observed for deaths not related to CE (Table 4).

## 4. Discussion

This multicenter retrospective study promoted by GETH-TC compared the incidence of CE and clinical outcomes between 461 AML patients with and without PCC undergoing allo-HCT with standard-dose PTCy-based prophylaxis. Despite a higher comorbidity burden, patients with baseline cardiac morbidity did not experience significantly increased rates of CE, NRM, or worse OS, supporting the feasibility of PTCy in this high-risk group when appropriate cardiovascular screening and monitoring are ensured.

PTCy is a well-established strategy for GVHD prophylaxis, particularly in haplo-HCT, but increasingly applied across donor types. Its immunomodulatory benefits have improved GVHD control and improve allo-HCT outcomes [1,2,3,4,20]. However, concerns remain regarding its potential cardiotoxicity. Cy-associated cardiac toxicity is known to be dose-dependent and clinically heterogeneous, including arrhythmias, heart failure, pericarditis, and acute coronary syndromes. Toxicity is largely driven by the metabolite acrolein, which induces oxidative stress, inflammation, calcium dysregulation, and endothelial injury [21,22,23]. In addition, in the context of allo-HCT, the concomitant exposition to additional cardiotoxic drugs together with early inflammation, endothelial activation and damage, and the onset of concomitant complications increases the risk of ECE in patients receiving PTCY [24,25,26].

In this study, ECE occurred in 11% of patients with PCC, with a median onset of 7 days post-HCT. Although proportions were higher than in those without cardiac disease, mortality directly attributed to ECE was similar between groups. Notably, the HCT process itself imposes substantial hemodynamic, inflammatory and endothelial stress that can precipitate both decompensation of pre-existing heart disease and de novo events in patients without PCC [27,28,29]. The guidelines also support the strategy we applied together with early post-transplant surveillance and cardio-oncology involvement [14], to detect subclinical injury and manage early events, thereby limiting ECE-related mortality despite higher event rates in those with prior cardiac disease.

LCE occurred less frequently than ECE; however, their clinical impact was nonetheless significant. Although not statistically significant, LCE were higher in number among patients with PCC. In addition, this delayed presentation may potentially reflect cumulative cardiovascular stress following allo-HCT, or a progressive evolution of the patient’s underlying cardiac disease. These findings lead to the hypothesis that, had patients with baseline cardiopathy been followed for a longer period, the incidence of LCEs might have continued to rise, and supporting the conduction of complementary studies in long-term survivors transplanted with PTCY-based prophylaxis.

Importantly, all LCE diagnosed in this group were de novo and differed from pre-transplant cardiac conditionings, thereby increasing the complexity of post-transplant cardiac surveillance and complicating the prediction of future cardiovascular events [13,14,30]. In addition, although LCE were less frequently fatal than ECE, they still impacted long-term outcomes, emphasizing the need for prolonged cardiovascular monitoring post-HCT and the establishment of dedicated long-term cardio-oncology services with dedicated follow-up in this high-risk population.

Although data on allo-HCT outcomes in patients with cardiac morbidity are limited, available evidence supports its use in carefully selected patients with adequate cardiovascular support. Previous studies have shown that cardiac conditions—such as arrhythmias or reduced left ventricular ejection fraction—can negatively affect transplant morbidity and outcomes [31,32]. Furthermore, some investigations have also compared the incidence of ECE between patients with and without pre-transplant cardiac morbidity. Interestingly, Yeh et al. reported a significant association between cardiac morbidity and higher ECE risk in a cohort of 585 adults undergoing HLA-matched allo-HCT [6]. In contrast, studies by Dulery et al., Pérez-Valencia et al., and our group (GETH-TC) found no such association in cohorts transplanted from different donor types [5,7,11]. These discrepancies may reflect stricter selection of allo-HCT candidates with cardiac disease at these centers, as clinically significant pre-transplant cardiac conditions remain key factors limiting eligibility [33,34].

These findings all together, globally underscore the critical role of specialized cardio-oncology programs in the multidisciplinary management need for appropriately evaluating and treating patients undergoing allo-HCT [14]. This multidisciplinary approach is essential not only for improving cardiovascular risk assessment, treating aggressively cardiovascular risk-factors, and implementing cardioprotective strategies, but also for ensuring the safe administration of PTCy in patients with baseline cardiac disease, considering its demonstrated effectiveness on GVHD prevention

Lastly, in our study, global outcomes were comparable between patients with and without cardiac morbidity. However, despite its low incidence, ECE was linked to higher mortality and NRM in both groups. One more time, these results underscore the importance of dedicated cardio-oncology programs in allo-HCT centers, implementing tools for identifying high-risk patients, and cardioprotective strategies to reduce such complications, regardless of PTCy use [14,35,36,37,38]. Preventive strategies such as pre-HCT echocardiography, ECG monitoring, careful selection of conditioning intensity, early recognition of cardiac symptoms, and the avoidance of additional cardiotoxic agents are key to minimizing cardiovascular risk. Moreover, it is essential to recognize that patients without PCC are also at risk for transplant-related cardiovascular stress, reinforcing the need for structured cardiovascular evaluation and dedicated follow-up protocols for all allo-HCT candidates.

A major strength of this study lies in its multicenter design, which included 17 transplant centers, the systematic pre-transplant cardiac evaluation applied across sites, and the use of robust statistical approaches such as PSM and IPW to minimize confounding and appropriately assess the safety of PTCY-based prophylaxis in patients with PCC considered eligible for allo-HCT. However, its retrospective nature, the variability in baseline cardiac conditions, the lack of data on patients with cardiac morbidity who were not eligible for allo-HCT, and the absence of details on cardiology-led monitoring strategies limit the study’s ability to offer practical guidance for improving supportive care in this patient group.

In addition, the small number of patients and events in the PCC group restricts the statistical power, especially for the subgroup analysis of ECE and LCE, and may account for some of the non-significant associations observed. These results should therefore be interpreted with caution and ideally confirmed in larger cohorts.

## 5. Conclusions

Our findings suggest that PTCy is safe and feasible in AML patients with PCC undergoing allo-HCT. While the incidence of ECE and LCE may be higher in this group, outcomes remain comparable when comprehensive cardiovascular evaluation and multidisciplinary care are provided. These results support the expansion of PTCy use in broader patient populations, guided by structured cardio-oncology collaboration.

## Figures and Tables

**Figure 1 cancers-17-03128-f001:**
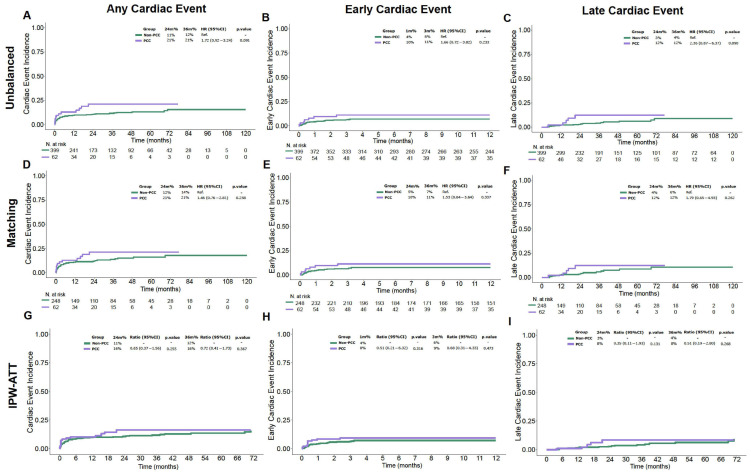
Cardiac Events in Patients with and without Cardiac Morbidity with Balanced Matching and IPW. IPW-ATT: inverse probability weighting; PCC: Pre-existing cardiac comorbidities. (**A**) Any cardiac event, unadjusted analysis; (**B**) Early cardiac event, unadjusted analysis; (**C**) Late cardiac event, unadjusted analysis; (**D**) Any cardiac event, propensity score matching (PSM); (**E**) Early cardiac event, PSM; (**F**) Late cardiac event, PSM; (**G**) Any cardiac event, inverse probability weighting (IPW); (**H**) Early cardiac event, IPW; (**I**) Late cardiac event, IPW.

**Figure 2 cancers-17-03128-f002:**
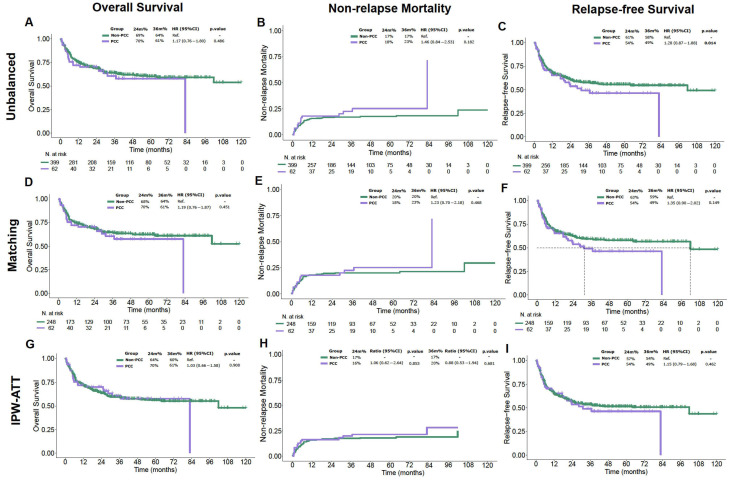
Overall efficacy outcomes after allo-HCT in patients with and without pre-transplant cardiac morbidity with unadjusted and adjusted analyses (PSM, IPW). IPW-ATT: inverse probability weighting; PCC: Pre-existing cardiac comorbidities. (**A**) Overall survival, unadjusted analysis; (**B**) Non-relapse mortality, unadjusted analysis; (**C**) Relapse-free survival, unadjusted analysis; (**D**) Overall survival, propensity score matching (PSM); (**E**) Non-relapse mortality, PSM; (**F**) Relapse-free survival, PSM; (**G**) Overall survival, inverse probability weighting (IPW); (**H**) Non-relapse mortality, IPW; (**I**) Relapse-free survival, IPW.

**Table 1 cancers-17-03128-t001:** Baseline characteristics of the study cohort.

	Overall N = 461 (%)	PCC N = 62 (%)	Non-PCC N = 399 (%)	*p* Value
**Median Age**	55 (43–63)	60 (48–66)	54 (42–63)	0.016
Sex				0.12
Male	236 (51)	38 (61)	198 (50)	
Female	225 (49)	24 (39)	201 (50)	
**Cardiovascular Risk Factors**				
HTA	399 (98)	19 (31)	76 (19)	0.05
Dyslipidemia	60 (13)	9 (15)	51 (13)	0.9
Mellitus Diabetes	36 (8)	7 (11)	29 (7)	0.4
Obesity	27 (6)	5 (8)	29 (7.3)	>0.9
**HCT-CI > 3**	73 (17)	15 (25)	58 (15)	0.2
Missing	23	1	22	
**Disease Status**				0.3
Complete Remission 1	345 (75)	42 (68)	303 (76)	
Complete Remission 2	69 (15)	13 (21)	56 (14)	
Refractory/Relapsed	47 (11)	7 (12)	40 (10)	
**Conditioning Intensity**				0.029
MAC	249 (54)	25 (40)	224 (56)	
RIC	212 (46)	37 (60)	175 (44)	
**Donor Type**				0.8
Matched Sibling Donor	86 (19)	10 (16)	76 (19)	
10/10 MUD	77 (17)	9 (15)	68 (17)	
7/8 MMUD	30 (6.5)	5 (8)	25 (6.3)	
Haploidentical	268 (58)	38 (61)	230 (58)	
**Day + 100 Cumulative Incidence of ECE**	-	11% (4.9–21%)	7% (4.8–9.9%)	0.53
**2-year Cumulative Incidence of LCE**	-	12% (4.4–25%)	2.9% (1.4–5.3%)	0.09
**Main Outcomes:**				
**2-year Overall Survival, % (IC95)**	-	70% (60–83%)	69% (64–73%)	0.5
**2-year Non-Relapse mortality, %**	-	17% (13–21%)	18% (9.5–29%)	0.2

ECE, early cardiac event; HCT-CI, hematopoietic cell transplantation–comorbidity index; HTA, hypertension; LCE, late cardiac event; MAC, myeloablative conditioning; MMUD, mismatched unrelated donor; MUD, matched unrelated donor; RIC, reduced-intensity conditioning.

**Table 2 cancers-17-03128-t002:** Baseline characteristics of patients with and without pre-existing cardiac morbidity before and after adjustment with Balanced Matching and inverse probability weighting.

	Unbalanced			IPW-ATT Balanced			PSM Balanced		
Characteristic	Non-PCC	PCC	SMD	Non-PCC	PCC	SMD	Non-PCC	PCC	SMD
	N = 399	N = 62		Weights Sum: 62	Weights Sum: 62		N = 248	N = 62	
**HTN**			0.3			0.01			0.1
No	323 (81%)	43 (69%)		43 (70%)	43 (69%)		178 (72%)	43 (69%)	
Yes	76 (19%)	19 (31%)		19 (30%)	19 (31%)		70 (28%)	19 (31%)	
Age	54 (42, 63)	60 (48, 66)		59 (49, 65)	60 (48, 66)	0.00	59 (50, 65)	60 (48, 66)	
**Conditioning intensity**			0.3			0.00			0.1
MAC	224 (56%)	25 (40%)		25 (41%)	25 (40%)		107 (43%)	25 (40%)	
RIC	175 (44%)	37 (60%)		37 (59%)	37 (60%)		141 (57%)	37 (60%)	
**Disease status**			0			0.01			0
RC	359 (90%)	55 (89%)		55 (89%)	55 (89%)		223 (90%)	55 (89%)	
R/R LMA	40 (10%)	7 (11%)		7 (11%)	7 (11%)		25 (10%)	7 (11%)	
**Donor**			0.1			0.02			0.1
MSD	76 (19%)	10 (16%)		10 (16%)	10 (16%)		45 (18%)	10 (16%)	
MUD	68 (17%)	9 (15%)		9 (15%)	9 (15%)		40 (16%)	9 (15%)	
MMUD	25 (6.3%)	5 (8.1%)		5 (8.6%)	5 (8.1%)		20 (8.1%)	5 (8.1%)	
HAPLO	230 (58%)	38 (61%)		37 (61%)	38 (61%)		143 (58%)	38 (61%)	
**HCT > 3**			0.2			0.01			0.1
No	333 (83%)	47 (76%)		47 (76%)	47 (76%)		193 (78%)	47 (76%)	
Yes	66 (17%)	15 (24%)		15 (24%)	15 (24%)		55 (22%)	15 (24%)	
**DLP**			0.1			0.01			0
No	348 (87%)	53 (85%)		53 (86%)	53 (85%)		210 (85%)	53 (85%)	
Yes	51 (13%)	9 (15%)		9 (14%)	9 (15%)		38 (15%)	9 (15%)	

DLP, dyslipidemia; HAPLO, haploidentical donor; HCT-CI, hematopoietic cell transplantation–comorbidity index; HTN, hypertension; IPW-ATT: inverse probability weighting; LMA R/R, relapsed/refractory acute myeloid leukemia; MAC, myeloablative conditioning; MMUD, mismatched unrelated donor; MSD, matched sibling donor; MUD, matched unrelated donor; PCC, pre-existing cardiac comorbidities; RC, complete remission; RIC, reduced-intensity conditioning. Difference indicates the standardized mean difference between groups; values closer to 0 indicate better balance.

**Table 3 cancers-17-03128-t003:** Comparison of cardiac events between patients with and without pre-existing cardiac morbidity.

Early Cardiac Events (ECE)	Total Events N = 35	PCC N = 7	Non-PCC N = 28	*p* Value
**ECE grade (CTCAE), n (%)**				
Grade I–II	9 (54.3)	3 (32.9)	16 (57.2)	0.064
Grade III–IV	12 (34.3)	2 (28.6)	10 (35.7)	
Grade V	4 (11.4)	2 (28.6)	2 (7.1)	
**ECE, n patients (%)**				
Arrythmia	11 (31.4)	3 (42.9)	8 (28.6)	0.528
Heart Failure	9 (25.7)	3 (42.9)	6 (21.4)	
Stroke/Ischemia	3 (8.6)	0	3 (10.7)	
Pericarditis/Pericardial Effusion	11 (31.4)	1 (14.3)	10 (35.7)	
Other (Cardiomiopathy)	1 (2.9)	0	1 (3.6)	
**Days to ECE, median (range)**	15 (8–49)	7 (3–29)	15 (8–51)	0.933
**Death due to CE**	4 (11.4)	5 (71.4)	2 (7.1)	0.171
**Death during follow-up**	17 (48.6)	2 (28.6)	12 (42.9)	0.228
Day +30 Mortality Rate	4 (11.4)	2 (28.6)	2 (7.1)	0.214
Day +60 Mortality Rate	6 (17.1)	2 (28.6)	4 (14.2)	0.889
**Main Outcomes:**				
**2-year Overall Survival, % (IC95)**	53.4 (35.6–68.3)	28.6 (4.1–61.2)	59.9 (39.3–75.4)	0.164
**2-year Non-Relapse mortality, %**	37.1 (21.3–53.0)	57.1 (12.1–86.2)	32.1 (15.8–49.8)	0.255
**Late Cardiac events (LCE)**	**Total events** **N = 26**	**PCC** **N = 5**	**Non-PCC** **N = 21**	***p*** **value**
**LCE grade (CTCAE), n (%)**				0.230
Grade I–II	17 (65.4)	5 (20.0)	12 (57.2)	
Grade III–IV	6 (23.1)	0	6 (28.59	
Grade V	2 (7.7)	0	2 (9.5)	
**LCE, n patients (%)**				
Arrythmia	5 (19.2)	0	5 (23.8)	0.391
Heart Failure	10 (38.5)	3 (60.0)	7 (33.5)	
Stroke/Ischemia	4 (15.4)	1 (20.0)	4 (19.0)	
Pericarditis/Pericardial Effusion	5 (19.2)	0	4 (19.0)	
Other	2 (7.7)	1 (20.0)	1 (4.8)	
**Days to LCE, median (range)**	437 (183–850)	469 (264–574)	383 (172–1087)	<0.001
Death due to CE	3 (11.5)	1 (20.0)	2 (9.5)	0.937
Death during follow-up	10 (38.5)	2 (40.0)	8 (38.1)	0.488
Day + 30 Mortality Rate	5 (19.2)	0	5	0.073
Day + 60 Mortality Rate	5 (19.2)	0	5	0.073
**Main Outcomes:**				
**2-year Overall Survival, % (IC95)**	76.2 (54.2–88.5)	100	70.2 (45.1–85.4)	0.887
**2-year Non-Relapse mortality, %**	3.8 (0.3–16.8)	0	4.8 (0.3–20.3)	0.149

Legend: Only cardiac events occurring in patients without disease relapse were accounted for the study conduction. De novo HTN was not accounted as CE. CE, cardiac event; CTCAE, Common Terminology Criteria for Adverse Events; ECE, early cardiac event; LCE, late cardiac event.

**Table 4 cancers-17-03128-t004:** Early and late cardiac events in patients with prior cardiac morbidity.

	ECE N = 7	LCE N = 5	*p* Value
**Prior Cardiac Morbidity**			0.76
Arrythmia	3 (42.8)	1 (20.0)	
Heart Failure (insuf ventric)	2 (28.5)	0	
Isquemia	0	1 (20.0)	
Pericardiac Disease	1 (14.3)	0	
Moderate/Server Valvopathy	1 (14.3)	3 (60.0)	
Other	0	0	
**CE, n patients (%)**			0.76
Arrythmia	3 (42.9)	0	
Heart Failure	3 (42.9)	3 (60.0)	
Stroke/Ischemia	0	1 (20.0)	
Pericarditis/Pericardial Effusion	1 (14.3)	0	
Other (Cardiomiopathy)	0	1 (20.0)	
**Newly diagnosed CE, n patients (%)**	4 (57.1)	5 (100)	0.20
**Descompensation, n patients (%)**	3 (42.9)	0	0.20

CE, Cardiac event; ECE, Early cardiac event; LCE, Late cardiac Event.

## Data Availability

Data sharing would be only considered after specific request.

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
