# Peer review of "Safety of Post-Transplant Cyclophosphamide-Based Prophylaxis in AML Patients with Pre-Existing Cardiac Morbidity Undergoing Allogeneic Hematopoietic Cell Transplantation"

_cancers, 2025, doi:10.3390/cancers17193128_

Round 1

Reviewer 1 Report

Comments and Suggestions for Authors

This multicenter retrospective study addresses the safety of post-transplant cyclophosphamide (PTCy) in AML patients with pre-existing cardiac morbidity undergoing allo-HCT. 

The study employs propensity score matching and inverse probability weighting to account for confounders.

Several methodological and presentation issues should be addressed.

Major Comments

1. Definition of Pre-Existing Cardiac Morbidity (PCC):

The definition includes both documented cardiac disease and LVEF <45%. While clinically reasonable, the heterogeneity within the PCC group (e.g., valvopathy, arrhythmias, heart failure) may mask subgroup-specific risks. Consider providing a subgroup analysis or at least a discussion of whether certain types of cardiac conditions, like LVEF<45%, are associated with higher risk.

2. Sample Size and Power:

The PCC group comprises only 62 patients, with 7 ECE and 5 LCE events. This limits the statistical power to detect significant differences, especially in subgroup analyses. The authors should acknowledge this limitation and caution against overinterpreting non-significant results.

3. Timing and Classification of Cardiac Events:

The classification of ECE (≤100 days) and LCE (>100 days) is standard, but the median time to LCE (~469 days in PCC group) suggests that longer follow-up might reveal more events. The authors should discuss whether the follow-up duration was sufficient to capture late cardiotoxicities.

4. Cause of Death and Competing Risks:

The analysis appropriately uses competing risk models. However, the cause of death attribution (e.g., directly due to CE vs. other causes) should be clarified, especially since CE-related mortality was low. Consider including a sensitivity analysis or cause-specific hazard model.

5. Missing Data and Imputation:

The authors used multiple imputation for missing data in the IPW model. The proportion and pattern of missingness should be reported (e.g., in a supplementary table), and the imputation method should be justified.

Minor Comments

Abstract:

1. The abstract states “11% vs. 7% ECE” and “8% vs. 5.3% LCE”, but the values in the results section (Table 1) are slightly different. Please ensure consistency.

2. Table 1:

(1)“HTA” should be spelled out as “Hypertension” for international readability.

(2)The last two rows “ Day +100 cumulative incidence of ECE” to “2-year non relapse mortality” were not the baseline characteristics. Please move the parts to the propriate place.

(3)”DLP” was not a commonly used abbreviation, and was not used in the table, please delete.

3. Table 2:

(1)Some percentages do not sum to 100% (e.g., ECE grade distribution). Please verify all calculations.

(2)The “Death due to CE” row in the ECE section appears inconsistent (PCC: 5/7? vs. Death during follow-up: 2/7?). Clarify the numerator/denominator.  Although P value was not significant, 5/7 death in PCC and 2/28 in non-PCC  should be discussed.

 (3) “main outcomes” and “Days to LCE” rows should be aligned properly.

4. References:

Some references are incomplete or formatted inconsistently (e.g., Ref 15). Please ensure all references follow the journal‘s style guide.

Author Response

Reviewer 1

First of all, we would like to thank Reviewer 1 for the time taken to review the manuscript, and the respectful and constructive contributions. We have carefully reviewed all the comments and addressed each of the concerns raised, with detailed responses provided below each comment.

Major Comments

Comment 1: Definition of Pre-Existing Cardiac Morbidity (PCC):

The definition includes both documented cardiac disease and LVEF <45%. While clinically reasonable, the heterogeneity within the PCC group (e.g., valvopathy, arrhythmias, heart failure) may mask subgroup-specific risks. Consider providing a subgroup analysis or at least a discussion of whether certain types of cardiac conditions, like LVEF<45%, are associated with higher risk.

Response 1: We thank the reviewer for this valuable comment. We agree that the PCC group is heterogeneous, as it includes patients with different cardiac conditions (e.g., valvopathy, arrhythmias, heart failure, reduced LVEF). However, the number of patients in each specific subgroup was too small to allow a meaningful statistical analysis, as this would have required separating them into independent categories with very limited power.

To clarify this point, we have revised the Methods section to explicitly state that, given the small number of patients within each condition, subgroup analyses were not feasible and therefore all patients fulfilling these criteria were analyzed collectively as PCC.

Lines 124-127: Given the small number of patients with each specific cardiac condition, subgroup analyses were not feasible, as separating them into independent categories would result in very limited statistical power. For this reason, all patients fulfilling these criteria were analyzed collectively as PCC.

Comment 2: Sample Size and Power:

The PCC group comprises only 62 patients, with 7 ECE and 5 LCE events. This limits the statistical power to detect significant differences, especially in subgroup analyses. The authors should acknowledge this limitation and caution against overinterpreting non-significant results.

Response 2: We agree with the reviewer that the relatively small number of patients and events in the PCC group limits the statistical power to detect significant differences, particularly in the subgroup analysis of ECE and LCE. This limitation has now been explicitly addressed in the Discussion section of the revised manuscript.

The revised text reads:

Lines 383-386: In addition, the small number of patients and events in the PCC group restricts the statistical power, especially for the subgroup analysis of ECE and LCE, and may account for some of the non-significant associations observed. These results should therefore be interpreted with caution and ideally confirmed in larger cohorts.

Comment 3: Timing and Classification of Cardiac Events:

The classification of ECE (≤100 days) and LCE (>100 days) is standard, but the median time to LCE (~469 days in PCC group) suggests that longer follow-up might reveal more events. The authors should discuss whether the follow-up duration was sufficient to capture late cardiotoxicities.

Thank you for the comment. We agree with the Reviewer that patient follow-up may limit the evaluation of “very-late” cardiac events, and the best methodology to address this hypothesis would be the conduction of specific studies in long-term survivors.

Notably, PTCy-based prophylaxis, especially outside the haplo-HCT setting, is still a relatively new approach for assessing transplant-related toxicities, and future studies exploring this aspect should be conducted in the coming years.

Based on the Reviewer’s comment, the following information has been included in the Discussion Section:

Lines 331-334: These findings lead to the hypothesis that, had patients with baseline cardiopathy been followed for a longer period, the incidence of LCEs might have continued to rise, and supporting the conduction of complementary studies in long-term survivors transplanted with PTCY-based prophylaxis.

Comment 4: Cause of Death and Competing Risks:

The analysis appropriately uses competing risk models. However, the cause of death attribution (e.g., directly due to CE vs. other causes) should be clarified, especially since CE-related mortality was low. Consider including a sensitivity analysis or cause-specific hazard model.

Response 4: We appreciate the reviewer’s suggestion. In our original analysis, the competing risk models were applied to cardiac events but not specifically to CE-related mortality. To address this comment, we performed additional cause-specific hazard models distinguishing deaths directly related to CE from deaths due to other causes. The results are presented in Supplementary Table 1.

In both the unadjusted and matched analyses, the hazard ratios for CE-related death were consistent with the overall competing risk findings (Unbalanced HR = 3.35, 95% CI 0.84–13.30, p = 0.086; Matched HR = 2.50, 95% CI 0.60–10.40, p = 0.197). No significant differences were observed for deaths not related to CE.

Model

Censor

Event

Competing Event

HR (95%CI)

p.value

Unbalanced

Competing risk

Alive at last follow-up

CE-related death

Death not related to CE

3.35

(0.84-13.30)

0.086

Global OS

Alive at last follow-up

Any death

-

1.17

(0.76-1.80)

0.486

CE OS

Alive at last follow-up

CE-related death

-

3.37

(0.84-13.5)

0.086

Other OS

Alive at last follow-up

Death not related to CE

-

1.07

(0.67-1.69)

0.781

Matching

Competing risk

Alive at last follow-up

CE-related death

Death not related to CE

2.50

(0.60-10.40)

0.197

Global OS

Alive at last follow-up

Any death

-

1.19

(0.76-1.87)

0.451

CE OS

Alive at last follow-up

CE-related death

-

2.48

(0.59-10.38)

0.195

Other OS

Alive at last follow-up

Death not related to CE

-

1.11

(0.69-1.79)

0.682

Lines 173-176: In addition to the main competing risk models, we performed cause-specific hazard models distinguishing deaths directly attributable to cardiac events from deaths due to other causes.

Lines 286-292: Among patients who experienced an ECE, 5 of 7 in the PCC group (71%) and 2 of 28 in the non-PCC group (7%) died directly due to the cardiac event. Although numbers were small and the difference did not reach statistical significance, this suggests a higher vulnerability among patients with pre-existing cardiac morbidity. To further address this, we performed cause-specific hazard models distinguishing CE-related deaths from deaths due to other causes (Supplementary Table 1), no significant differences were observed for deaths not related to CE.

Comment 5: Missing Data and Imputation:

The authors used multiple imputation for missing data in the IPW model. The proportion and pattern of missingness should be reported (e.g., in a supplementary table), and the imputation method should be justified.

Response 5: We thank the reviewer for this helpful suggestion. We have added a supplementary table (Supplementary Table 3) summarizing the proportion and distribution of missing data across covariates used in the IPW model, as well as a comparison between imputed and non-imputed datasets. As shown, the proportion of missing values was very low (<5% for all variables).

In the revised manuscript, the description of missing data handling in the Statistical Methods section now reads:

Lines 190-194: “Missing data in the covariates used for the IPW model were minimal (<5% for all variables, see Supplementary Table 3). Missingness was assumed to be completely at random. We applied multivariate imputation by chained equations (MICE), a flexible approach that handles both continuous and categorical variables. Estimates obtained from imputed datasets were consistent with complete case analyses.”

These additions clarify the extent and handling of missing data and support the robustness of our findings

Characteristic

Imputed

No imputed

p-value2

N = 4611

N = 4611

HTA1

>0.9

    No

366 (79%)

365 (79%)

    Si

95 (21%)

95 (21%)

    Missing

0

1

Age

>0.9

    Mean (SD)

52 (14)

52 (14)

    Median (Q1, Q3)

55 (43, 63)

55 (43, 63)

    Min, Max

17, 79

17, 79

Intensity1

>0.9

    MAC

249 (54%)

249 (54%)

    RIC

212 (46%)

212 (46%)

Status2

>0.9

    RC

414 (90%)

414 (90%)

    R/R LMA

47 (10%)

47 (10%)

Donor

>0.9

    MSD

86 (19%)

86 (19%)

    MUD

77 (17%)

77 (17%)

    MMUD

30 (6.5%)

30 (6.5%)

    HAPLO

268 (58%)

268 (58%)

HCT_mas3

0.8

    No

380 (82%)

365 (83%)

    Si

81 (18%)

73 (17%)

    Missing

0

23

DLP1

>0.9

    No

401 (87%)

399 (87%)

    Si

60 (13%)

60 (13%)

    Missing

0

2

Minor Comments

Abstract:

Comment 1: The abstract states “11% vs. 7% ECE” and “8% vs. 5.3% LCE”, but the values in the results section (Table 1) are slightly different. Please ensure consistency.

Response 1: We thank the reviewer for noting this inconsistency. The proportions have now been corrected in the Abstract to match the values reported in the Results section (Table 1). Specifically, for ECE the rates are 7/62 (11.3%) in the PCC group and 28/399 (7.0%) in the non-PCC group, and for LCE the rates are 5/62 (8.1%) and 21/399 (5.3%), respectively. The Abstract has been updated to ensure consistency throughout the manuscript.

Comment 2: Table 1:

(1)“HTA” should be spelled out as “Hypertension” for international readability.

(2)The last two rows “ Day +100 cumulative incidence of ECE” to “2-year non relapse mortality” were not the baseline characteristics. Please move the parts to the propriate place.

(3)”DLP” was not a commonly used abbreviation, and was not used in the table, please delete.

Response 2:

  • We thank the reviewer for pointing this out. “HTA” has been corrected to “Hypertension” for international readability.
  • We acknowledge the reviewer’s concern. To clarify, the last rows presenting cumulative incidence of ECE, overall survival (OS), and non-relapse mortality (NRM) were intentionally separated and presented in bold to distinguish them from baseline characteristics. This structure was chosen to align with the way results are presented in the manuscript text, where “Cumulative incidence of ECE,” “OS,” and “NRM” are described in the Main Outcomes section.
  • We agree with the reviewer. The abbreviation “DLP” has been removed, as it was not used consistently in the table.

Comment 3: Table 2:

(1)Some percentages do not sum to 100% (e.g., ECE grade distribution). Please verify all calculations.

(2)The “Death due to CE” row in the ECE section appears inconsistent (PCC: 5/7? vs. Death during follow-up: 2/7?). Clarify the numerator/denominator.  Although P value was not significant, 5/7 death in PCC and 2/28 in non-PCC  should be discussed.

 (3) “main outcomes” and “Days to LCE” rows should be aligned properly.

Response 3:

  • We thank the reviewer for noting this. All percentages in Table 2 have been carefully re-verified and corrected where necessary to ensure they sum to 100%.
  • We appreciate the reviewer’s observation regarding the “Death due to CE” row. To clarify, “Death due to CE” refers specifically to deaths directly attributable to the cardiac event, whereas “Death during follow-up” reflects overall mortality among patients who experienced an ECE, regardless of the cause. We have revised the table and its footnote to make this distinction clearer. Also, we mentioned it in results and in limitations.

Lines 286-287: Among patients who experienced an ECE, 5 of 7 in the PCC group (71%) and 2 of 28 in the non-PCC group (7%) died directly due to the cardiac event

Lines 383-386: In addition, the small number of patients and events in the PCC group restricts the statistical power, especially for the subgroup analysis of ECE and LCE, and may account for some of the non-significant associations observed. These results should therefore be interpreted with caution and ideally confirmed in larger cohorts.

  • We thank the reviewer for pointing out the formatting issue. The “Main outcomes” and “Days to LCE” rows have now been properly aligned to improve clarity and consistency in Table 2.

Comment 4: References:

Some references are incomplete or formatted inconsistently (e.g., Ref 15). Please ensure all references follow the journal‘s style guide.

Response 4: We thank the reviewer for pointing this out. All references have been carefully reviewed and revised to ensure completeness and consistency according to the journal’s style guide. In particular, the formatting of Ref. 15 and any other inconsistencies have been corrected.

Reviewer 2 Report

Comments and Suggestions for Authors

The manuscript by Torrent et al. describes a safety study of the cardiotoxicity of cyclophosphamide (PTCy) in acute myeloid leukemia (AML) patients receiving allogeneic hematopoietic cell transplantation (allo-HCT). The authors carried out a retrospective, multicenter study comprising 17 institutions and 461 AML patients, and analyzed the cumulative incidence, overall survival (OS), non-relapse mortality (NRM) and incidence of cardiac events (CE). They showed that there is no significant differences in CE incidence, OS, or NRM between the risk groups, and their conclusion supported the safe use of PTCy in AML patients with pre-existing cardiac morbidity.

Comments and suggestions:

  1. Line 203-206, "…the difference was not statistically significant when compared to patients without…" Did the authors perform power analysis to determine whether this was due to the sample size?
  2. Line 222-223, "2 (28.6%) patients with PCC and 2 (7.1%) patients without died directly due to ECE (grade 5)", like the previous remark, the single-digit number is too small to achieve enough statistical power.
  3. Table 2, it would be better to include the comparison where early and late CEs are combined for the sake of better statistical power.
  4. Figure 1-2, please increase the font size for the inset text as they are too small to be legible.
  5. Figure 2C,F, why do the curves for the Non-PCC group (green) look almost identical? The "N at risk" seems to suggest different proportions especially at the early time points.

Author Response

Comment 1: Line 203-206, "…the difference was not statistically significant when compared to patients without…" Did the authors perform power analysis to determine whether this was due to the sample size?

Response 1: We thank the reviewer for this insightful comment. We did not perform a formal power analysis for this study. As acknowledged in the revised Discussion section, the relatively small sample size—particularly within the PCC group—may have limited the statistical power to detect significant differences. This limitation has now been explicitly addressed in the manuscript.

The revised text reads:

Lines 383-386: In addition, the small number of patients and events in the PCC group restricts the statistical power, especially for the subgroup analysis of ECE and LCE, and may account for some of the non-significant associations observed. These results should therefore be interpreted with caution and ideally confirmed in larger cohorts.

Comment 2: Line 222-223, "2 (28.6%) patients with PCC and 2 (7.1%) patients without died directly due to ECE (grade 5)", like the previous remark, the single-digit number is too small to achieve enough statistical power.

Response 2: We fully agree with the reviewer that the small number of events limits the statistical power to detect meaningful differences, particularly for outcomes such as death directly attributable to ECE. We have clarified this point in the revised Discussion (lines xx-xxx), emphasizing that these results should be interpreted with caution and ideally confirmed in larger studies.

Comment 3: Table 2, it would be better to include the comparison where early and late CEs are combined for the sake of better statistical power.

Response 3: We appreciate the reviewer’s suggestion. We have considered combining early and late cardiac events to increase statistical power. However, given the different pathophysiological mechanisms and timing, we preferred to present ECE and LCE separately, in line with previous studies. Nonetheless, to address the reviewer’s concern, we have highlighted in the Discussion that the limited number of events in each subgroup restricts the statistical power (lines 383-386).

Comment 4: Figure 1-2, please increase the font size for the inset text as they are too small to be legible.

Response 4: We thank the reviewer for this observation. The font size of the inset text in Figures 1 and 2 has been increased in the revised version to improve readability.

Comment 5: Figure 2C,F, why do the curves for the Non-PCC group (green) look almost identical? The "N at risk" seems to suggest different proportions especially at the early time points.

Response 5: We thank the reviewer for this observation. Panels C and F show the Non-PCC group before (unbalanced cohort, Panel C) and after propensity-score matching (Panel F). While the absolute numbers at risk differ (Unbalanced: 399 → 256 → 185; Matched: 248 → 159 → 119 at 0, 12 and 24 months), the proportions of patients remaining at risk are very similar (12 months: 64.2% vs 64.1%; 24 months: 46.4% vs 48.0%). This explains why the survival curves look almost the same. At the same time, survival estimates at specific time points and the corresponding hazard ratios differ slightly, reflecting the effect of matching.

Round 2

Reviewer 2 Report

Comments and Suggestions for Authors

The authors have addressed my concerns and revised the manuscript. I do not have other questions.